# Clinical PD-1/PD-L1 Blockades in Combination Therapies for Lymphomas

**DOI:** 10.3390/cancers15225399

**Published:** 2023-11-14

**Authors:** Hiroo Katsuya, Junji Suzumiya, Shinya Kimura

**Affiliations:** 1Division of Hematology, Respiratory Medicine and Oncology, Department of Internal Medicine, Faculty of Medicine, Saga University, Saga 849-8501, Japan; 2Department of Hematology, Koga Community Hospital, Yaizu 425-0088, Japan; j-suzumiya@sunkohkai.or.jp

**Keywords:** lymphoma, PD-1/PD-L1 blockade, combination therapy with immunotherapy

## Abstract

**Simple Summary:**

Immunotherapy using antibodies against programmed cell death protein 1 (PD-1) or its ligand PD-L1 can restore host antitumor immunity in many types of cancer. Hence, PD-1/PD-L1 blockade therapy has become the standard treatment for various cancer types in the last decade. However, apart from classic Hodgkin lymphoma and primary mediastinal B-cell lymphoma, PD-1/PD-L1 blockade therapy has shown limited efficacy in other lymphomas. To address this gap, several clinical trials of combination therapies with PD-1/PD-L1 inhibitors have been recently conducted or are underway in both frontline and relapsed/refractory settings. Here, we comprehensively review these clinical studies of combination therapies for lymphomas and discuss their outcomes in the hope of providing a perspective to develop novel therapeutic approaches for combination therapy.

**Abstract:**

Immunotherapy with the programmed cell death protein 1 (PD-1)/PD-1 ligand (PD-L1) blockade has revolutionized the treatment of advanced solid cancers. However, these clinical benefits have been limited to cases of malignant lymphomas, showing promising results for only classic Hodgkin lymphoma (cHL) and primary mediastinal B-cell lymphoma (PMBCL). To bring clinical benefits to more patients with lymphoma, numerous combination therapies involving PD-1/PD-L1 blockade have been tested in clinical trials in both frontline and relapsed/refractory settings. This article reviews the current landscape of combination therapies with PD-1/PD-L1 blockade for lymphoma and discusses the potential therapeutic approaches. An interim analysis of a phase 3 study demonstrated increased progression-free survival with nivolumab combination therapy over the current frontline treatment in patients with advanced-stage cHL. The results of combination therapies for aggressive B-cell lymphomas, except for PMBCL, have been disappointing. Several clinical trials of combined PD-1/PD-L1 blockade and Bruton’s tyrosine kinase inhibitors are exploring its efficacy in patients with chronic lymphocytic leukemia (CLL) with Richter transformation. Several T-cell lymphoma subtypes respond to PD-1/PD-L1 blockade monotherapy. Further clinical trials are underway to investigate appropriate combination regimens with PD-1/PD-L1 blockade, especially for cHL, CLL with Richter transformation, and T-cell lymphoma, in both frontline and relapsed/refractory settings.

## 1. Introduction

Programmed cell death protein 1 (PD-1) is an important receptor that controls overt immune responses to maintain immunological homeostasis [1,2]. In the tumor environment, PD-1 ligands (PD-L1 and PD-L2) expressed on tumor cells and tumor-associated macrophages (TAMs) bind to PD-1 expressed on activated tumor-infiltrating lymphocytes (TILs), leading to suppression of the T-cell response to tumor cells. This interaction in the tumor environment promotes tumor evasion, resulting in tumor growth. Inhibiting these interactions using monoclonal antibodies against PD-1 or PD-L1 can restore the antitumor immunity of the host. Therefore, immunotherapy with the PD-1/PD-L1 blockade has revolutionized the treatment of many cancers in the last decade.

Mature lymphoid neoplasms are divided into two categories: B-cell and T/NK-cell lymphoid proliferations/lymphomas [3]. Classic Hodgkin lymphoma (cHL) is a neoplasm derived from B-cells, which are composed of a small number of Hodgkin/Reed-Sternberg (HRS) cells in the background of reactive inflammatory cells, including T-lymphocytes, plasma cells, granulocytes, and histiocytes. HRS cells express PD-L1, while PD-1 is markedly elevated in TILs [4]. The frequencies of 9p24.1 alterations, where the genes for PD-L1 and PD-L2 are located, are 56% copy gain, 36% amplification, and 5% polysomy in patients with cHL, all of which are associated with PD-L1 protein expression [5,6]. PD-L1 is also expressed by TAMs, which have both tumor-promoting and immunosuppressive functions. The abundant PD-L1-expressing TAMs colocalize with PD-L1-expressing HRS cells in the tumor microenvironment and are frequently in contact with CD4^+^ T cells [7]. In contrast to cHL, diffuse large B-cell lymphoma (DLBCL) varies in the expression of PD-1 and PD-L1. Notably, 9p24.1 copy gain is observed in only 6–19% of patients with DLBCL [8,9,10] but in more than 50% of patients with primary mediastinal B-cell lymphoma (PMBCL) and primary central nervous system lymphoma (PCNSL) [11]. Rearrangement of 9p24.1 is also observed in 20% of PMBCL cases and is associated with PD-L1 and PD-L2 overexpression [12]. These statistics provide a biological rationale for the treatment of patients with cHL and PMBCL with PD-1/PD-L1 blockade.

Nivolumab and pembrolizumab, both of which are monoclonal antibodies against PD-1, have proven effective in patients with relapsed/refractory (R/R) cHL. In a phase 2 clinical trial (CheckMate 205) examining nivolumab treatment in 243 patients with R/R cHL who relapsed after autologous stem cell transplantation (ASCT) [13], the overall response rate (ORR) and complete response rate (CRR) were 69% and 16%, respectively. At the 5-year follow-up, the median progression-free survival (PFS) was 15.1 months, and the median overall survival (OS) was not reached [14]. In the phase 2 KEYNOTE-087 study on pembrolizumab in 210 patients with R/R cHL, the ORR and CRR were 69% and 22%, respectively [15]. For R/R PMBCL, phase 1b (KEYNOTE-013) and phase 2 (KEYNOTE-170) studies have assessed the efficacy and safety of pembrolizumab treatment [16]. The ORRs were 48.0%, including a CRR of 33%, among the 21 patients in KEYNOTE-013 and 45%, including a CRR of 13%, among the 53 patients in KEYNOTE-170. Additionally, treatment-related adverse events (AEs) were observed in 24% of patients in KEYNOTE-013 and 23% of patients in KEYNOTE-170. There were no treatment-related deaths. These studies led to the FDA approval of both nivolumab and pembrolizumab for the treatment of R/R cHL and PMBCL.

Immunotherapy with PD-1/PD-L1 blockade can be extraordinarily effective in patients with various cancers, but clinical benefits have been limited to a minority of patients. The underlying reasons for the low efficacy of PD-1/PD-L1 blockade monotherapies may include defects in antigen presentation, non-inflamed tumor microenvironment, and genetic factors. Several immunogenic chemotherapies such as cyclophosphamide and oxaliplatin induce tumor CD8^+^ T-cell infiltration and delay cancer growth in clinically relevant mouse models [17]. These drugs can make resistant tumors sensitive to checkpoint blockade therapy. Further, chemotherapy may generate immunogenic neoantigens in dying tumor cells through their capacity to induce immunogenic cell death [18]. Fragmented immunogenic proteins generated by cisplatin-treated apoptotic tumor cells act as non-muted neoantigens and elicit effective multi-specific CD4^+^ and CD8^+^ T-cell responses, and checkpoint blockade therapies induce a further increase of the frequency of such T-cell responses [19]. Moreover, intracellular expressed PD-L1 protein governs the cellular response to DNA damage by engaging in the binding and stability of the mRNA encoding DNA damage-related genes [20,21]. The emergence of small molecule inhibitors targeting PD-1/PD-L1 has demonstrated promising cellular inhibitory efficacy, potentially mitigating the limitations associated with monoclonal antibodies [22,23].

Thus, a combination approach with PD-L/PD-L1 blockade and chemotherapy may be beneficial in terms of enhancing the immune system’s response to the tumor, potentially leading to a more effective anti-cancer treatment. To further improve the clinical outcomes of various types of lymphoma, numerous clinical trials of combination therapies with PD-1/PD-L1 inhibitors have been recently conducted or are underway in both frontline and relapsed/refractory settings. Here, we provide a summary of combination therapies involving PD-1/PD-L1 blockade and discuss potential approaches to combination therapy.

## 2. Classic Hodgkin Lymphoma

### 2.1. Frontline Setting

Nivolumab or pembrolizumab combined with conventional chemotherapies have shown promising clinical activity in patients with early- or advanced-stage cHL. Frontline checkpoint blockade in cHL was recently reported in four studies. In advanced stages III to IV and IIB with unfavorable risk factors, Cohort D of a phase 2 trial (CheckMate 205) assessed nivolumab monotherapy followed by nivolumab plus doxorubicin, vinblastine, and dacarbazine (AVD) for newly diagnosed cHL (Table 1) [24]. The patients received nivolumab monotherapy for four cycles, followed by six cycles of nivolumab-AVD chemotherapy. As a result, the ORR was 84%, with a CRR of 67%, and the PFS at 9 months was 92%. Notably, patients with higher PD-L1 expression showed more favorable responses to nivolumab plus AVD treatment. Grade 3–4 AEs were observed in 59% of patients, including neutropenia (49%) and febrile neutropenia (10%). The most frequent immune-related AE (irAE) was hypothyroidism, which was observed in eight patients (16%, all grade 1 to 2).

The complete metabolic response (CMR) rate and safety of sequential pembrolizumab and AVD chemotherapy were assessed in patients newly diagnosed with cHL (Table 1) [25]. The patients were treated with three cycles of pembrolizumab monotherapy, followed by four to six cycles of AVD, depending on the stage and bulk of tumors. All patients showed CMR after two cycles of AVD and maintained their response at the end of treatment. At a median follow-up of 22.5 months, PFS and OS were both 100% [26]. High response rates were observed in patients with all PD-L1 expression levels, including those with low levels, and grade 1–2 and grade 3–4 irAEs were observed in 94% and 6% of patients, respectively.

The results of the interim analysis of a phase 3 trial (SWOG S1826) comparing nivolumab-AVD with brentuximab vedotin (BV)-AVD in patients with newly diagnosed advanced cHL (stage III to IV) have been reported (Table 1) [27]. Briefly, patients received six cycles of nivolumab-AVD on days 1 and 15 or BV-AVD on days 1 and 15 with granulocyte colony-stimulating factor (G-CSF) neutropenia prophylaxis. The PFS in the nivolumab-AVD arm was superior to that in the BV-AVD arm (1-year PFS, 94% vs. 86%, respectively; hazard ratio, 0.48; *p* = 0.0005). Grade 3–4 hematological AEs were observed in 48% and 31% of patients in the nivolumab-AVD and BV-AVD arms, respectively. Hypothyroidism and hyperthyroidism occurred more frequently in the nivolumab-AVD arm, whereas peripheral neuropathy was more common in the BV-AVD arm. Follow-up analysis is ongoing to assess the long-term PFS, safety, and OS of these patients. The nivolumab-AVD regimen may become the new standard chemotherapeutic regimen for patients with advanced-stage cHL.

A randomized phase 2 German Hodgkin Study Group NIVAHL trial for newly diagnosed cHL with early-stage unfavorable cHL compared concomitant therapy with four cycles of nivolumab-AVD to sequential therapy with four cycles of nivolumab, two cycles of nivolumab-AVD, and two cycles of AVD, both consolidated by 30 Gy involved-site radiotherapy (ISRT) (Table 1) [28]. Among 101 patients, 90% of those receiving concomitant therapy and 94% of those receiving sequential therapy achieved CR after treatment. After a median follow-up of 41 months, the 3-year PFS rates were 100% and 98% for concomitant and sequential therapies, respectively, and the 3-year OS rate was 100% in both treatment groups [29]. For patients newly diagnosed with stage I/II cHL, two phase 2 trials evaluating the efficacy of adding nivolumab to the standard treatment are currently ongoing (NCT03233347 and NCT03712202). A phase 3 clinical trial is also underway to compare the PFS of a standard chemotherapy approach versus immunotherapy (BV and nivolumab) in newly diagnosed patients with stage I/II cHL who have a rapid early response, as determined by positron emission tomography (PET) after two cycles of ABVD (NCT05675410) (Appendix A). Adding immunotherapy to the standard treatment regimen may not only increase survival but also result in fewer short-term and long-term side effects.

**Table 1 cancers-15-05399-t001:** Treatment outcomes of PD-1/PD-L1 blockade combination therapies in classic Hodgkin lymphoma.

	Disease Condition	Combination Therapy	Phase	Number of Patients	ORR (%)	CRR (%)	PFS (%)	OS (%)	AE Grade 3–4	Reference
Frontline	Early-stage unfavorable	Nivo-AVD × 4 cycles	2	51	100	90	100% at 3 y	100% at 3 y	11%	[28,29]
		Nivo × 4 cycles, nivo-AVD × 2 cycles, followed by AVD × 2 cycles		50	98	94	98% at 3 y	100% at 3 y	7%	
	Early-stage unfavorable, or advanced-stage	Nivo × 4 cycles, followed by AVD × 6 cycles	2	51	84	67	92% at 9 m	98% at 9 m	59%	[24]
	Early-stage unfavorable, or advanced-stage	Pem × 3 cycles, followed by AVD × 4–6 cycles	2	30	100	100	100% at 22.5 m	100% at 22.5 m	6%	[25,26]
	Advanced-stage	Nivo-AVD × 6 cycles	3	489	NA	NA	94% at 1 y	99% at 1 y	NA	[27]
		vs. BV-AVD × 6 cycles		487	NA	NA	86% at 1 y	98% at 1 y	NA	
Relapsed/ Refractory	First salvage	Nivo-BV × 4 cycles	1/2	91	85	67	77% at 3 y	93% at 3 y	33%	[30,31]
	Post-ASCT consolidation	Nivo-BV × 8 cycles	2	59	NA	NA	92% at 2 y	98% at 2 y	NA	[32]
	First salvage	Nivo × 3 cycles, followed by nivo × 3 cycles or nivo-ICE. Pts. in CR/PR proceed to ASCT.	2	43	93	91	72% at 2 y	95% at 2 y	NA	[33]
	First salvage	Pem-ICE × 3 cycles, followed by pem × 1 cycle	2	37	97	87	87 at 2 y	95 at 2 y	NA	[34]
	Post multi-salvage	Nivo-ipilimumab	1b	31	74	23	not reached at 1 y	NA	29%	[35]

Abbreviations: nivo, nivolumab; pem, pembrolizumab; AVD, doxorubicin, vinblastine, and dacarbazine; ASCT, autologous stem cell transplantation; ICE, ifosfamide, carboplatin, and etoposide; BV, brentuximab vedotin; ORR, overall response rate; CRR, complete response rate; PR, partial response; PFS, progression-free survival; OS, overall survival; AE, adverse event; NA, not available.

### 2.2. Relapsed/Refractory Setting

While the standard of care for patients with R/R cHL is salvage chemotherapy followed by ASCT, monotherapy with nivolumab or pembrolizumab has proven to be effective in these patients as well. Moreover, combination therapies involving PD-1 blockade have also shown clinical efficacy in this setting. A phase 1/2 study evaluated BV combined with nivolumab as the first salvage therapy in patients with R/R cHL (Table 1) [30,31]. Briefly, these patients received up to four cycles of combination therapy with BV on day 1 and nivolumab on day 8 of the first cycle. For cycles 2–4, BV and nivolumab were administered on day 1. The ORR was 85% with a CRR of 67%. After a median follow-up of 34.3 months, the PFS and OS at 3 years were 77% and 93%, respectively. The PFS at 3 years was 90% for patients with relapsed disease and 61% for patients with primary refractory disease. Among the 67 patients (74%) who underwent ASCT directly after treatment with BV plus nivolumab, the PFS at 3 years was 91%. Grade 3–4 AEs were observed in 33% of patients, and irAEs of any grade were observed in 18% of patients, including pneumonitis in four patients (4%) and maculopapular rash, increased aspartate aminotransferase (AST), diarrhea, and Guillain–Barre syndrome, each occurring in one patient (1%).

A phase 2 study sequentially evaluated the safety and efficacy of nivolumab combined with BV as a post-ASCT consolidation therapy for high-risk R/R cHL (Table 1) [32]. High-risk features included primary refractory cHL, relapsed cHL within 1 year of completing the initial therapy, extranodal involvement or B symptoms at relapse, or more than one salvage regimen used prior to ASCT. The proportions of patients with prior BV and anti-PD1 exposure were 51% and 42%, respectively. The nivolumab plus BV therapy was initiated between days 30 and 60 after ASCT and administered every 21 days for up to eight cycles. At a median follow-up of 29.9 months, the PFS rates at 18 and 24 months were 94% and 92%, respectively, meeting the primary endpoint of this study. The most common grade 3–4 AEs were neutropenia (31%) and pneumonitis (7%), while the most common irAEs of any grade were elevated AST levels (15%), pneumonitis (14%), rash (12%), hyperbilirubinemia (7%), and hypothyroidism (7%). The combination of nivolumab and BV may be an attractive therapy for post-ASCT consolidation and awaits further clinical trials.

Another phase 2 trial evaluated nivolumab monotherapy followed by nivolumab combined with ifosfamide, carboplatin, and etoposide (ICE) as the first salvage therapy in R/R cHL (Table 1) [33]. These patients received nivolumab monotherapy every 2 weeks for three cycles, and PET-CT was performed to assess the disease response. Patients in the CR or PR groups received another three cycles of nivolumab, followed by another PET-CT scan. Patients who achieved CR after six cycles of nivolumab underwent ASCT. Patients with progressive disease (PD) after three or six cycles of nivolumab received two cycles of nivolumab plus ICE therapy. Patients with stable disease (SD) after three cycles of nivolumab could proceed with an additional three cycles of nivolumab or nivolumab plus ICE. Patients who achieved PR or CR after nivolumab plus ICE could undergo ASCT. Thirty-four and nine patients received nivolumab alone and nivolumab plus ICE, respectively. After nivolumab monotherapy, the ORR was 81% with a CRR of 71%. At the end of the therapy, the ORR and CRR were 93% and 91%, respectively. The PFS and OS at 2 years in all treated patients were 72% and 95%, respectively. Among the nine patients who received nivolumab plus ICE, no unacceptable toxicities were noted. The most common grade 3 to 4 nivolumab plus ICE-related AEs were neutropenia and hypophosphatemia, each observed in two patients, and febrile neutropenia, colitis, syncope, and leukopenia, each observed in one patient. Moreover, a phase 2 trial of pembrolizumab combined with ICE therapy evaluated the CRR using PET-CT in patients with R/R cHL who were scheduled to undergo ASCT (Table 1) [34]. The patients received two cycles of pembrolizumab with ICE every 21 days, followed by stem cell collection, and one cycle of pembrolizumab monotherapy followed by PET/CT. The ORR and CRR were 97.3% and 86.5%, respectively. The PFS and OS at 2 years were 87.2% and 95.1%, respectively. These two studies suggest that PD-1 blockade combined with ICE salvage therapy may be well-tolerated and effective in patients with R/R cHL eligible for ASCT.

Cytotoxic T lymphocyte-associated protein 4 (CTLA-4) is an immunogenic checkpoint that downregulates immune responses. Nivolumab in combination with ipilimumab, a CTLA-4 antibody, has shown a clinically meaningful improvement in the survival of patients with several cancers. A phase 1 study (CheckMate 039) explored the safety and efficacy of nivolumab plus ipilimumab combination therapy in patients with R/R cHL, non-Hodgkin lymphoma, or multiple myeloma (Table 1) [35]. In patients with cHL, the ORR was 74%, including 23% who achieved CR, a result that was not significantly different from that expected with nivolumab monotherapy. In addition, 29% of the patients experienced grade 3–4 AEs, with 8% of the patients ceasing the study treatment because of AEs. Thus, the combination of nivolumab and ipilimumab may not be a useful treatment option in patients with R/R cHL, unlike several solid cancers.

Several more clinical trials for cHL are exploring the efficacy and safety of PD-1/PD-L1 inhibitors in combination with other therapies, including cytotoxic agents (AVD, ICE, and bendamustine) and BV (Appendix A).

## 3. B-Cell Lymphoma

### 3.1. Diffuse Large B-Cell Lymphoma (DLBCL)

Previous clinical trials of frontline PD-1/PD-L1 blockade combination therapy have not proven to be effective in patients with DLBCL. A phase 1b/2 clinical trial assessed the safety and efficacy of atezolizumab, a humanized IgG1 monoclonal antibody targeting PD-L1, in combination with R-CHOP (rituximab, cyclophosphamide, doxorubicin, vincristine, and prednisolone) followed by consolidation of atezolizumab monotherapy in patients with newly diagnosed DLBCL (Table 2) [36]. In this trial, the CRR and ORR were 78% and 88%, respectively. The PFS and OS at 3 years were 77.4% and 87.2%, respectively. Grade 3–4 AEs were observed in 76% of patients, and irAEs of any grade were observed in 40.5% of patients. Furthermore, pembrolizumab with R-CHOP was tested in the same setting in patients with DLBCL (Table 2) [37]. The ORR was 90%, with a CRR of 77%, and the PFS at 2 years was 83%. Grade 3–4 AEs and irAEs of any grade were observed in 43% and 13% of patients, respectively. In addition, a phase 2 trial investigated durvalumab, a human IgGκ monoclonal antibody against PD-L1, combined with R-CHOP or lenalidomide plus R-CHOP, in patients with newly diagnosed high-risk DLBCL (Table 2) [38]. High-risk DLBCL was defined as Ann Arbor stage III to IV or stage II with bulky disease and intermediate-high or high disease risk (international prognostic index (IPI) ≥ 3 or National Comprehensive Cancer Network-IPI ≥ 4). Of the 46 patients, 43 received durvalumab plus R-CHOP and 3 received lenalidomide plus R-CHOP. Among the 37 patients evaluated for efficacy in the durvalumab plus R-CHOP arm, the ORR was 73%, including a CRR of 54%, and the PFS at 1 year was 67.7%. Grade 3–4 AEs and irAEs of any grade were observed in 83.7% and 60.5% of patients, respectively. These three clinical trials of combination therapy with PD-1/PD-L1 blockade showed no greater efficacy or safety benefits than R-CHOP alone, and additional AEs should be carefully considered.

Combination therapies involving PD-1/PD-L1 blockade have also been tested in patients with R/R DLBCL. A cohort of 11 patients with DLBCL were treated with nivolumab plus ipilimumab in a phase 1 study (CheckMate 039) (Table 2) [35]. In these patients, the ORR was 18%, including a CRR of 9%, which was an insufficient response rate compared with other combination therapies for R/R DLBCL. Furthermore, a high number of patients experienced AE of grades 3 to 4, with 8% ceasing the study treatment as a result, as mentioned in the section on cHL. Another phase 1b study assessed the safety and efficacy of a combination of atezolizumab and obinutuzumab in patients with R/R DLBCL or follicular lymphoma (FL) (Table 2) [39]. The ORR was 17%, including a CRR of 4%, in patients with R/R DLBCL, which did not show a significant improvement in the response rate compared to that of obinutuzumab monotherapy. A phase 1b study assessed the safety and efficacy of treatment with a combination of rituximab, atezolizumab, and polatuzumab vedotin in 21 patients with R/R DLBCL (Table 2) [40]. The ORR was 25% with a CRR of 13%. A different phase 1b study assessed the safety and efficacy of treatment with a combination of atezolizumab, obinutuzumab, and venetoclax in 58 patients with R/R DLBCL (Table 2) [41]. In this case, the ORR was 24%, including a CRR of 18%, and grade 3–4 AEs were observed in 94% of patients. Finally, another phase 1b study evaluated the safety and efficacy of blinatumomab and pembrolizumab combination therapy in 31 patients with R/R DLBCL (Table 2) [42]. Grade 3–4 AEs occurred in 94% of these patients, and the most frequent serious AEs attributed to blinatumomab were nervous system disorders (35%). The ORR was 30%, and the median duration of response in responders was 176.5 days. Although numerous clinical trials of combination therapy with PD-1/PD-L1 inhibitors have been conducted for R/R DLBCL, no significant improvement in overall activity has been observed to date.

Several clinical trials for DLBCL are exploring the efficacy and safety of PD-1/PD-L1 inhibitors in combination with other agents, including conventional regimens (R-CHOP and rituximab plus gemcitabine-oxaliplatin), Bruton’s tyrosine kinase (BTK) inhibitors, and chimeric antigen receptor T-cell (CAR-T) therapies (Appendix A). CAR-T therapy is a relatively new option with the potential to achieve durable remission and possibly cure patients with multiply relapsed DLBCL. However, 60–70% of the patients relapsed in the first year after infusion [43]. The addition of PD-1 blockade to CAR-T therapy may overcome the inhibitory effect of PD-1 expression and may result in enhanced efficacy of CAR-T therapy and eventually tumor suppression (NCT05385263, NCT05659628).

**Table 2 cancers-15-05399-t002:** Treatment outcomes of PD-1/PD-L1 blockade combination therapies in B-cell lymphoma.

Lymphoma Subtype	Disease Condition	Combination Therapy	Phase	Number of Patients	ORR (%)	CR (%)	PFS	OS	AE Grade 3–4	Reference
DLBCL	Frontline	Atezo-R-CHOP × 6–8 cycles	1b/2	40	88	78	77% at 3 y	87% at 3 y	76%	[36]
		Pem-R-CHOP × 6 cycles	2	30	90	77	83% at 2 y	84% at 2 y	43%	[37]
		Durva-R-CHOP × 6–8 cycles, followed by consolidation with durva × 12 cycles	2	37	73	54	68% at 1 y	NA	84%	[38]
	Relapsed/ refractory	Nivo-ipilimumab	1	11	18	9	NA	NA	29%	[35]
		G × 1 cycle, followed by atezo-G × 7 cycles, and then atezo only	1b	23	17	4	Median, 3 m	NA	61%	[39]
		Atezo-rituximab-polatuzumab vedotin × 6 cycles	1b	21	25	13	NA	NA	24%	[40]
		Atezo-G-venetoclax × 8 cycles, followed by atezo-venetoclax × 16 cycles	2	58	24	18	NA	NA	84%	[41]
		Pem-blinatumomab	1b	31	30	NA	NA	NA	94%	[42]
PMBCL	Relapsed/ refractory	Nivo-brentuximab vedotin	1/2	29	74	40	56% at 2 y	76% at 2 y	53%	[44,45]
FL	Frontline	Nivo × 4 cycles, followed by nivo × 4 cycles, and nivo maintenance × 12 cycles, or followed by nivo-rituximab × 4 cycles, and nivo-rituximab maintenance × 8 cycles	2	39	92	54	72% at 1 y	96% at 1 y	NA	[46]
		Atezo-G-bendamustine, followed by maintenance with atezo-G	1b/2	40	85	75	81% at 3 y	89% at 3 y	80%	[47]
	Relapsed/ refractory	G × 1 cycle, followed by atezo-G × 7 cycles, and then atezolizumab only	1b	26	54	23	Median, 9.5 m	NA	61%	[39]
		Atezo-G-polatuzumab vedotin × 6 cycles, followed by atezo-G maintenance	1b	10	50	20	NA	NA	62%	[40]
		Atezo-G-lenalidomide × 6 cycles, followed by atezo-G-lenalidomide maintenance	1b/2	38	78	72	68% at 3 y	90% at 3 y	84%	[48]
		Atezo-G-venetoclax × 8 cycles, followed by atezo-venetoclax × 16 cycles	2	58	54	30	NA	NA	71%	[49]
CLL	Relapsed/ refractory	Pem-ibrutinib-fludarabine	2	10	100	10	NA	NA	NA	Not published
	Richter transformation	Nivo-ibrutinib	2	24	42	33	NA	Median, 13 m	12%	[50]
		Nivo-ibrutinib-blinatumomab	2	9	22	11	Median, 1.9 m	Median, 11.5 m	NA	Not published

Abbreviations: DLBCL, diffuse large B-cell lymphoma; PMBCL, primary mediastinal B-cell lymphoma; FL, follicular lymphoma; CLL, chronic lymphocytic leukemia; nivo, nivolumab; pem, pembrolizumab; durva, durvalumab; atezo, atezolizumab; G, obinutuzumab; R-CHOP, rituximab, cyclophosphamide, doxorubicin, vincristine, and prednisolone; ORR, overall response rate; CR, complete response; PFS, progression-free survival; OS, overall survival; AE, adverse event; NA, not available.

### 3.2. Primary Mediastinal B-Cell Lymphoma (PMBCL)

PD-1 blockade monotherapy has proven to be effective for the treatment of R/R PMBCL. Although PMBCL shares many clinical and molecular features with cHL, CD30 is more variably expressed compared to that in cHL [51]. A phase 2 study of BV monotherapy for CD30-positive PMBCL demonstrated an ORR of 13% [52]. Furthermore, the safety and efficacy of nivolumab combined with BV were evaluated in a phase 1/2 study (CheckMate 436) for patients with CD30-positive non-Hodgkin lymphoma (NHL), defined as a CD30 expression level of 1% or greater in the tumor or tumor-infiltrating lymphocytes by immunohistochemistry (Table 2) [44]. The patients were treated with nivolumab and BV once every 3 weeks. In 29 patients with PMBCL, the ORR was 74% with a CRR of 40% [45]. The PFS and OS rates at 24 months were 55.5% and 75.5%, respectively. Grade 3–4 AEs were observed in 53.3% of the patients. The most common irAEs were diarrhea and rash. The high response rate supports the use of nivolumab combined with BV as a salvage therapy for ASCT. Some clinical trials have explored the efficacy and safety of PD-1/PD-L1 blockade in addition to DA-EPOCH (dose-adjusted cyclophosphamide, doxorubicin, etoposide, vincristine, and prednisolone), BV plus R-CHP, and CAR-T therapy (Appendix A). In the frontline setting, a phase 2 clinical trial is ongoing to assess the efficacy of BV and nivolumab followed by R-CHP (rituximab, cyclophosphamide, doxorubicin, and prednisolone) in patients with newly diagnosed PMBCL (NCT04745949).

### 3.3. Primary Central Nervous System Lymphoma (PCNSL) and Primary Testicular Lymphoma (PTL)

Copy number alterations of chromosome 9p24.1 have been reported in more than 50% of PCNSL and PTL cases [10]. A retrospective study reported that five patients (100%) receiving nivolumab for R/R PCNSL and CNS relapse of PTL showed clinical responses [53]; however, this result could not be confirmed in a prospective study including 47 patients with PCNSL and 19 with CNS relapse of PTL, according to the available results posted on clinicaltrials.gov in 2023 (NCT02857426). Pembrolizumab monotherapy was also tested in 41 and 9 patients with R/R PCNSL and primary vitreoretinal tumors, respectively. The ORR was 26%, and the median PFS was 2.6 months [54]. Two BTK inhibitors, ibrutinib and tirabrutinib, have been studied in phase 1 and 2 trials as a monotherapy in R/R PCNSL and resulted in dramatic responses but poor durability [55,56]. Further studies are exploring the efficacy and safety of PD-1/PD-L1 blockade in combination therapies with BTK inhibitors (NCT03770416 and NCT04462328) (Appendix A).

### 3.4. Follicular Lymphoma

PD-1 expression levels are low (6–11%) in FL [57]. Although PD-1^+^ TILs in FL may receive inhibitory signals from PD-L1 expressed on histocytes and regulatory T-cells [58,59], the prognostic significance of this interaction in FL is unclear. A phase 2 study of nivolumab monotherapy for R/R FL showed limited activity, with an ORR of 4% and a median PFS of 2.2 months [60]. In a frontline setting, a phase 2 study of nivolumab combined with rituximab assessed their efficacy and safety as a first-line therapy in patients with grade 1–3a FL (Table 2) [46]. Briefly, patients received biweekly nivolumab induction for four cycles. Patients with CR received four further cycles of nivolumab monotherapy, followed by 12 cycles of maintenance with nivolumab on a 4-weekly basis. Patients with PR, SD, and asymptomatic or minor PD received nivolumab plus rituximab biweekly for four cycles, followed by maintenance with the combination on a 4-weekly basis for eight cycles. The ORR was 92% with a CRR of 54%, and the PFS and OS rates at 12 months were 72% and 96%, respectively. Grade 3–4 AEs were observed in 41% of patients, and grade 3–4 irAEs were infrequent. Follow-up is ongoing to assess the long-term PFS and safety of this combination. A phase 1b/2 trial assessed the safety and efficacy of induction therapy with atezolizumab in addition to obinutuzumab and bendamustine, followed by maintenance with atezolizumab–obinutuzumab for ≤24 months in patients with untreated grade 1–3a FL (Table 2) [47]. The ORR was 85%, including a CRR of 75%, and the PFS and OS at 3 years were 80.9% and 89.3%, respectively. Notably, grade 5 AEs were reported in five patients (12.5%), which were caused by pneumonia, sudden death, cardiac arrest (following severe immune-mediated myocarditis and bronchiolitis obliterans), adenocarcinoma (likely primary site: gastrointestinal tract/biliary origin), and progressive multifocal leukoencephalopathy. The addition of atezolizumab to obinutuzumab and bendamustine increased the risk of significant AEs and is not recommended in patients with untreated FL.

In the R/R setting, a phase 1b study assessed the safety and efficacy of a combination of atezolizumab and obinutuzumab in 26 patients with FL (Table 2) [39]. The ORR was 54% with a CRR of 23%, and the median PFS was 9.5 months. Notably, the addition of atezolizumab did not improve PFS in patients with R/R FL. Another phase 1b study assessed the safety and efficacy of adding atezolizumab to obinutuzumab and polatuzumab vedotin in 10 patients with R/R FL (Table 2) [40]. The ORR was 50% with a CRR of 20%. Grade 3–5 AEs were observed in 62% of patients, including two deaths, with 31% of patients developing a serious AE. The addition of atezolizumab to obinutuzumab and lenalidomide in phase 1b/2 resulted in an ORR of 78%, including a CRR of 72%. Grades 3–5 AEs were observed in 84% of patients and included two deaths (Table 2) [48]. In addition, a phase 2 study of the addition of atezolizumab to obinutuzumab and venetoclax showed an ORR of 54%, a CRR of 30%, and grade 3–4 AEs in 71% of patients (Table 2) [49]. Therefore, the addition of PD-1/PD-L1 inhibitors does not offer a significant clinical benefit in the R/R setting and carries with it an increased risk of clinically significant AEs.

### 3.5. Chronic Lymphocytic Leukemia (CLL)

CLL treatment has dramatically changed with the development of small-molecule inhibitors targeting BTK, B-cell lymphoma 2 (BCL2), and phosphatidylinositol-3 kinase (PI3K). Unfortunately, PD-1 blockade has not proven to be an effective target therapy for patients with CLL. Exhausted effector T-cells in patients with CLL overexpress PD-1 and are too defective to form immune synapses with tumor cells [61,62,63]. However, flow cytometric analysis showed weak expression of PD-1 and no expression of PD-L1/L2 in tumor cells [64]. A phase 2 study of pembrolizumab monotherapy for relapsed CLL failed to demonstrate a clinical benefit; the ORR was 0% in 16 patients, and the median PFS was 3 months [65]. The addition of pembrolizumab to ibrutinib and fludarabine was evaluated in 10 high-risk patients and in those with R/R CLL (Table 2). High risk was defined as those with a deletion of 17p, *TP53* mutation, *NOTCH1* mutation, *SF3B1* mutation, *MYC* aberration, or complex cytogenetics. The ORR was 100%, including a CRR of 10%; however, serious AEs were observed in 73% of patients according to the available results posted on clinicaltrials.gov in 2023 (NCT03204188).

In contrast to those with regular CLL, significant efficacy of PD-1 blockade has been reported in patients with Richter transformation. A phase 2 study of pembrolizumab monotherapy in nine patients with Richter transformation showed an ORR of 44%, including a CRR of 11%. The median PFS and OS were 5.4 months and 10.7 months, respectively [65]. Analyses of pretreatment tumor specimens revealed an increased expression of PD-L1 and a trend of increased expression of PD-1 in the tumor microenvironment in patients with confirmed responses. Another phase 2 clinical trial assessed the efficacy of combined nivolumab and ibrutinib in patients with CLL with or without Richter transformation (Table 2) [50]. In the 24 patients with Richter transformation, the ORR was 42%, including a CRR of 33%, and the median PFS was 13 months. Furthermore, 3 of the 13 (23%) patients who had prior exposure to BTK inhibitors showed a response, and 5 of the 11 patients with del(17p) and/or *TP53* mutations responded to combination therapy (all CMR). Grade 3–4 AEs were observed in 12% of the patients, and 17% of the patients had irAEs of any grade. Furthermore, a phase 2 study of the addition of blinatumomab to nivolumab and ibrutinib for patients with Richter transformation resulted in an ORR of 22%, including a CRR of 11%, and a median PFS of 1.9 months, according to the available results posted on clinicaltrials.gov in 2023 (NCT03121534). Given the limited therapeutic options for patients developing Richter transformation, the combination of nivolumab and ibrutinib may be a potential regimen for these patients. A few ongoing studies are investigating PD-1/PD-L1 inhibitors in combination with other BTK inhibitors and a BCL2 inhibitor (NCT05388006 and NCT02846623) (Appendix A).

## 4. T-Cell Lymphoma

Generally, the prognoses of most patients with T-cell lymphomas are inferior to those of patients with B-cell lymphomas or cHL, especially in cases of adult T-cell leukemia/lymphoma (ATL), hepatosplenic T-cell lymphoma, and aggressive NK-cell leukemia [66,67]. Neither tumor cells nor TILs of T-cell lymphoma generally express PD-1/PD-L1, with the exceptions of those in nodal T-follicular helper cell lymphoma, angioimmunoblastic-type (nTFHL-AI) [64], ATL [68], cutaneous T-cell lymphoma [69], and extranodal NK/T cell lymphoma [70]. nTFHL-AI is derived from T follicular helper cells and typically expresses PD-1 [71]. PD-L1 expression in tumor cells has been reportedly associated with a poor prognosis in ATL [68]. A phase 2 study of nivolumab monotherapy in 12 patients with R/R peripheral T-cell lymphomas (PTCL) included 6 patients with nTFHL-AI and 3 with PTCL, not otherwise specified (NOS) [72]. The ORR was 33%, with two cases of CR and two of PR. However, the median PFS was short (2.7 months), and the median OS was 6.7 months. Hyperprogressive disease occurred in four patients, three of whom had nTFHL-AI. A phase 2 study of pembrolizumab in 24 patients with R/R mycosis fungoides reported an ORR of 38%, including a CRR of 4% [73]. Two phase 2 studies on avelumab (PD-L1 inhibitor) and sintilimab (PD-1 inhibitor) in patients with R/R extranodal NK/T-cell lymphoma showed ORRs of 38% and 75%, respectively [74,75]. In a phase 2 trial of nivolumab for ATL, rapid disease progression was observed in all three patients after a single dose of nivolumab, and the clinical trial was subsequently terminated [76]. Another phase 2 trial of nivolumab in patients with aggressive R/R ATL in Japan did not show such a clinical course [77], and the study is currently ongoing (UMIN000020601). While the etiology of rapid progression following PD-1 blockade in nTFHL-AI and ATL remains enigmatic, it may arise from the disruption of a PD-1/PD-L1-induced tumor-suppressive effect. Wartewig et al. demonstrated that PD-1 has a tumor suppressor function in T-cell lymphoma in a mouse model [78]. PD-L1-expressing cells in the tumor microenvironment suppress PD-1^+^ ATL cells, and PD-1 blockade neutralizes this suppressive function. PD-1 blockade also altered the expression of ATL-promoting growth factors such as IL-2 and IL-10 [79]. Further studies are needed to understand the underlying mechanism to find risk factors for rapid progression and appropriate prevention strategies.

Although PD-1 blockade can lead to responses in several subtypes of T-cell lymphoma, further clinical trials are needed to assess its efficacy and safety. More studies examining combination therapies with PD-1/PD-L1 blockade are exploring the efficacy and safety of cytotoxic agents (NCT03586999 and NCT0359899) and histone deacetylase inhibitors (NCT03278782), among others (Appendix A).

## 5. PD-1/PD-L1 Expression in Lymphomas

The effects of PD-1/PD-L1 blocking agents on lymphoma are thought to be related to several complex factors. Therefore, we believe that the therapeutic effect of PD-1 antibody drugs does not simply correlate with PD-L1 expression in lymphoma cells. PD-L1 is expressed in almost all Hodgkin cells [80]. In cases of DLBCL, PD-L1 expression was found in 8.9% of DLBCL NOS cases (97 of 1091), 15.6% of EBV-positive DLBCL in the elderly (14 of 90), 30.8% of T-cell/histiocyte-rich large B-cell lymphoma cases (8 of 26), 5.6% of cases of other iatrogenic immunodeficiency-associated lymphoproliferative disorders (1 of 18), 45.5% of intravascular large B-cell lymphoma cases (5 of 11), and 42.9% of PMBCL cases (3 of 7) [8]. In cases of FL, PD-1 expression in lymphoma cells is low (6–11%) [57]. Indeed, another report showed that PD-L1 expression in FL lymphoma cells was found in only 1 of 173 cases [81]. In CLL, PD-1 expression is restricted to the proliferating central cells. However, in cases of Richter transformation, strong PD-1 expression was observed in the scattered background-reactive small T-cells in all 15 cases, and PD-1 expression was also observed in neoplastic large B-cells in 12/15 (80%) cases [82]. In T-cell lymphoma, the expression of PD-1/PD-L is restricted to cases of nTFHL-AI [60], ATL [48], cutaneous T-cell lymphoma [49], and extranodal NK/T cell lymphoma [61]. In the future, it will be necessary to determine the treatment strategy based on the expression patterns of PD-1 and PD-L1 in individual cases, as is performed for solid tumors.

## 6. Conclusions

In this article, we reviewed clinical studies on combination therapies with PD-1/PD-L1 inhibitors for the treatment of lymphomas. Apart from cHL and PMBCL, combination therapies with PD-1/PD-L1 blockade have shown limited efficacy in other lymphomas. Interim analysis of the SWOG S1826 phase 3 trial for advanced-stage cHL demonstrated an increased PFS in patients treated with nivolumab-AVD compared to those treated with the current frontline treatment (BV-AVD), indicating that nivolumab-AVD may become a new standard of care for patients with advanced-stage cHL. To date, although numerous clinical trials of combination therapy with PD-1/PD-L1 blockade have explored its efficacy in patients with DLBCL NOS, they have generally shown disappointing results. Furthermore, the addition of PD-1/PD-L1 blockade to standard care in patients with FL increased their risk of clinically significant AEs. The clinical benefits of PD-1 blockade in patients with T-cell lymphoma are restricted to those with nTFHL-AI, mycosis fungoides, and extranodal NK/T-cell lymphoma. Owing to the rarity of the disease, there are no randomized trials of combination therapies with PD-1/PD-L1 blockade in T-cell lymphoma and only small studies such as phase 2 and non-randomized trials are currently ongoing.

Further clinical research is necessary to explore the appropriate combination regimens and predictive biomarkers of response in patients treated with PD-1/PD-L1 blockade.

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
