# Peer review of "Clinical PD-1/PD-L1 Blockades in Combination Therapies for Lymphomas"

_cancers, 2023, doi:10.3390/cancers15225399_

Round 1
Reviewer 1 Report
Comments and Suggestions for Authors
Immune checkpoint inhibitors (PD1/PD-L1 antibodies) have transformed outcomes for subgroups of patients suffering with certain types of solid malignancies, but their efficacy in most lymphomas has been negligible. Notable exceptions to this paradigm include classical Hodgkin (cHL) and primary mediastinal B cell lymphomas (PMBL).
The manuscript (MS) from Katsuya and colleagues, reviews clinical experience with PD-1/PD-L1 antibodies in lymphoma. From a clinical perspective, this review appears to be comprehensive, and as far as I can tell, up to date. The review is well written although some of the data might be better summarized in tables for each specific disease.
What is lacking almost completely is any reference to the underlying biology. This is disappointing. No biological basis is provided for possible rational combinations.
As a consequence there are innumerable empirical clinical trials assessing different combinations in different clinical situations.
For example (one of many) lines 340-342 – which if any of these trials is worth pursuing? Why combine PD-1/PD-L1 inhibitors with any of these molecules?
An updated table of all the current studies involving ICI in lymphoma might be helpful, with updated current status – several appear to have been either terminated or closed to recruitment.
The results in Richter transformation are perhaps overstated and there is a general lack of enthusiasm for ICI in this condition. For example:-
Line 373 NCT03121534 – this trial was terminated 17FEB2023
Line 376 NCT03892044 – this trial is not recruiting
Line 386 - hyper-progressive disease in T-cell malignancies is very interesting but no biological basis for this is given. Why is rapid progression seen in ATL?
Reviewer 2 Report
Comments and Suggestions for Authors
In this research, the authors reviewed the status of Clinical PD-1/PD-L1 Blockades in Combination Therapies for Lymphomas. In my opinion, the current version of this manuscript fits the scope of Cancers and could be accepted after major revision.
My specific comments are in detail listed below:
1. In Lines 36-45, the current development of the PD-L1 pathway and its relevant new functions should be revealed, especially how PD-L1 affects the DNA repair process. Some references should be added to this part including 10.1016/j.ijbiomac.2022.10.167.
2. Some minor mistakes exist in the references. The authors should carefully check the format of the reference style.
3. In the introduction, the newly developed or discovered PD-L1 small molecular inhibitors rather than only PD-L1 or PD-1 antibodies could be added. Some references should be added to this part including 10.1016/j.jconrel.2022.11.004.
4. Some references are out of date (before 2010). Some new recent references may be better.
5. In conclusion, the clinical transformation barriers of PD-1/PD-L1 Blockades in Combination Therapies for Lymphomas should be better out-looked.
Round 2
Reviewer 2 Report
Comments and Suggestions for Authors
The current version of this manuscript could be accepted.
Author Response
Thank you for for your comment.